# Stringent Nonpharmaceutical Interventions Are Crucial for Curbing COVID-19 Transmission in the Course of Vaccination: A Case Study of South and Southeast Asian Countries

**DOI:** 10.3390/healthcare9101292

**Published:** 2021-09-29

**Authors:** Zebin Zhao, Xin Li, Feng Liu, Rui Jin, Chunfeng Ma, Bo Huang, Adan Wu, Xiaowei Nie

**Affiliations:** 1Key Laboratory of Remote Sensing of Gansu Province, Northwest Institute of Eco-Environment and Resources, Chinese Academy of Sciences, Lanzhou 730000, China; zhaozebin@lzb.ac.cn (Z.Z.); liufeng@lzb.ac.cn (F.L.); jinrui@lzb.ac.cn (R.J.); machf@lzb.ac.cn (C.M.); wuadan@lzb.ac.cn (A.W.); 2College of Resources and Environment, University of Chinese Academy of Sciences, Beijing 100049, China; 3National Tibetan Plateau Data Center, State Key Laboratory of Tibetan Plateau Earth System, Resources and Environment (TPESRE), Institute of Tibetan Plateau Research, Chinese Academy of Sciences, Beijing 100101, China; 4CAS Center for Excellence in Tibetan Plateau Earth Sciences, Chinese Academy of Sciences, Beijing 100101, China; 5Department of Geography and Resource Management, The Chinese University of Hong Kong, Hong Kong 999077, China; bohuang@cuhk.edu.hk; 6The Alliance of International Science Organizations, Beijing 100101, China

**Keywords:** COVID-19, improved SIRV, parameter estimation, South and Southeast Asia, scenario prediction, nonpharmaceutical interventions

## Abstract

The ongoing spread of coronavirus disease 2019 (COVID-19) in most South and Southeast Asian countries has led to severe health and economic impacts. Evaluating the performance of nonpharmaceutical interventions in reducing the number of daily new cases is essential for policy designs. Analysis of the growth rate of daily new cases indicates that the value (5.47%) decreased significantly after nonpharmaceutical interventions were adopted (1.85%). Vaccinations failed to significantly reduce the growth rates, which were 0.67% before vaccination and 2.44% and 2.05% after 14 and 28 d of vaccination, respectively. Stringent nonpharmaceutical interventions have been loosened after vaccination drives in most countries. To predict the spread of COVID-19 and clarify the implications to adjust nonpharmaceutical interventions, we build a susceptible–infected–recovered–vaccinated (SIRV) model with a nonpharmaceutical intervention module and Metropolis–Hastings sampling in three scenarios (optimistic, neutral, and pessimistic). The daily new cases are expected to decrease rapidly or increase with a flatter curve with stronger nonpharmaceutical interventions, and the peak date is expected to occur earlier (5–20 d) with minimum infections. These findings demonstrate that adopting stringent nonpharmaceutical interventions is the key to alleviating the spread of COVID-19 before attaining worldwide herd immunity.

## 1. Introduction

Since the beginning of December 2019, coronavirus disease 2019 (COVID-19) has had disruptive effects worldwide. The unprecedented COVID-19 pandemic, as a public health crisis prompted by a hitherto unknown virus, has threatened the health and safety of millions of people, and critically affected the international political and socioeconomic structure [1,2,3]. The first wave of the pandemic in 2021 had a stronger trend than the previous wave; the growth curve of the pandemic in late February 2021 was steeper than that in in previous months. The largest number of new cases in a single day worldwide exceeded 900,000, nearly half of which pertained to India for several consecutive days. The rebound of the epidemic in India and mutation of the virus (e.g., SARS-CoV-2 Delta variant) have posed severe threats to countries in South and Southeast Asia [4] and hampered the global process of curbing the pandemic. The cumulative number of confirmed cases worldwide has exceeded 200 million, and the impending new wave is expected to have an irreversible and ongoing impact on future efforts [5,6].

With the worldwide introduction of emergency vaccination in December 2020, the spread of COVID-19 was expected to slow [7], and the goal of “restarting the world” reached a turning point. However, the rebound of the pandemic in late February 2021 severely limited this process. Asia again emerged as a key region for risk with a large number of daily new cases due to the emergence of the highly transmissible Delta variant, especially in India [8]. The COVID-19 rebound in India led to the aggravation of the pandemic in many countries in South and Southeast Asia owing to the geographical proximity, the large number of slums with densely populated regions, frequent personnel exchanges [9], weak contact tracing, failure to inspect and quarantine at the borders, and limited medical supplies [10]. The inferior healthcare facilities in these regions and substantial variations in the population density, income levels, sociocultural aspects, and public health infrastructure will render South and Southeast Asia highly vulnerable if the number of severe cases of COVID-19 increase [4,11,12]. In addition, most of these countries have not achieved herd immunity, and their vaccination rates are extremely low [13]. Singapore, Bhutan, and Maldives have the highest full-vaccination rates of 74%, 61.5%, and 55%, respectively; however, Cambodia, Malaysia, Sri Lanka, and Brunei have rates of only 46.3%, 39.3%, 25.8%, and 15.1%, respectively, and less than 15% of the total population of 12 other countries (e.g., Afghanistan, Bangladesh, India, Indonesia, Laos, Myanmar, Nepal, Pakistan, the Philippines, Thailand, Timor, and Vietnam) has been vaccinated, as of August 21, 2021. The minimum requirement for herd immunity is for more than 60% of the total population to be vaccinated (or more than 120 doses of vaccine per 100 people) [14,15,16,17]. Furthermore, the authorized vaccines are insufficient to subdue the mutated virus [18]. The effectiveness of one dose of the Pfizer vaccine and ChAdOx1 nCoV-19 vaccine is reported to be 48.7% and 30.7% against the Alpha and Delta variants, respectively [19]. Vaccine breakthrough infections likely cannot alleviate the COVID-19 transmission dynamics in the long-term. Pandemic prevention and control in most South and Southeast Asian countries remain challenging owing to the impending new wave of the pandemic, shortage of vaccines, increased infectivity of the mutated virus (e.g., SARS-CoV-2 Delta and Lambda variants), and shorter incubation periods (2–4 d). Moreover, the rebound of the epidemic in India and virus mutation have severely damaged the economic system, which had not entirely recovered from the first wave. Despite its large pharmaceutical industry, India’s goal of providing a large number of vaccines globally appears difficult to achieve. This aspect is expected to seriously affect the vaccination status in more than 60 countries worldwide and delay the process of herd immunity attainment in developing countries.

Although complete containment of the spread of COVID-19 depends on vaccination, strong nonpharmaceutical interventions can prevent large-scale spread of the virus. Different nonpharmaceutical interventions have different impacts on curbing the spread of COVID-19. The use of face masks in public let to a greater decline in daily new case growth rates compared with that in the regions that did not issue similar mandates [20,21]. Major nonpharmaceutical interventions, such as lockdowns, had a larger effect on reducing the COVID-19 transmission compared with self-isolation, social distancing, ban of public events and school closure [22]. Rigorous contact tracing, positive testing, and other measures can also effectively control the spread of COVID-19 [20]. Several researchers have examined the impacts of nonpharmaceutical interventions. An et al. compared six interventions and indicated that the mask mandate was associated with low infection rates in the short term, and its early adoption enhanced long-term efficacy. Measures, such as lockdowns, international travel bans, public event bans, and schools and restaurant closures exhibited a lower efficacy in containing COVID-19 transmission [23]. Haug et al. demonstrated that gathering cancellations, school closures, and border restrictions are the most effective measures to curb the spread of the virus [24]. Thus, nonpharmaceutical interventions have promoted the prevention and control of the spread of COVID-19. Institutional infrastructure, enduring policy instruments, and acceleration of the adoption of nonpharmaceutical intervention policies can efficiently alleviate the spread of COVID-19 [25,26,27,28]. For example, China adopted strict restrictions, and other measures began to be introduced on 23 January 2020, which led to a decline in the number of daily new cases. Hong Kong and Singapore adopted strict pre-emptive measures prior to the outbreak of COVID-19 and successfully contained the first wave of infections [20]. However, premature relaxation of containment measures led to further outbreaks of COVID-19. Singapore is a representative case of mismanagement on this front. The country saw remarkable progress in the early stages of the crisis. However, after witnessing the initial success, the government relaxed its stringent stance, and the country experienced a surge of infections [25]. At present, only a combination of enhanced vaccination measures and nonpharmaceutical interventions can ensure the health and well-being of citizens [29,30,31]. However, most South Asian, especially, Southeast Asian countries, find it challenging to promptly implement effective interventions because they are developing countries with a relatively low socioeconomic status, large number of slums, high population density, rigid social stratification, scarce medical resources, different cultural characteristics, and limited number of COVID-19 vaccines [32].

By evaluating the effectiveness of specific prevention and control measures for daily new cases in South and Southeast Asian countries, feasible safety measures in the post-pandemic era of large-scale vaccination can be identified. The evaluation of the growth rate of daily new cases can provide reference for adjusting intervention measures. Moreover, accurate forecasts of the spread of COVID-19 can provide scientific reference for vaccination and nonpharmaceutical interventions. Accurate predictions of the spread of COVID-19 in South and Southeast Asian countries under three scenarios (optimistic: 1.5 times the current intervention intensity, neutral: current intervention intensity, and pessimistic: 50% of the current intervention intensity) can be used as a reference for intervention measure implementation in these countries and global pandemic status examination, especially to promote epidemic prevention and control in developing countries. In addition, COVID-19 spread prediction in India can promote global vaccine capacity assessment, understanding the herd immunity status and decision making for the appropriate time for the world to restart safely.

## 2. Materials and Methods

### 2.1. Data

Data used in this study included pandemic diagnosis data, vaccination data, and prevention and control measures (detailed rules). Verified COVID-19 data were obtained from Johns Hopkins University (https://github.com/CSSEGISandData/COVID-19, accessed on 21 August 2021) [33]. Vaccine data were obtained from Our World in Data (https://github.com/owid/COVID-19-data, accessed on 21 August 2021) [13]. The time series data of global pandemic prevention and control measures (detailed rules) were obtained from the Oxford COVID-19 government response tracking system (https://github.com/OxCGRT/COVID-policy-tracker, accessed on 21 August 2021) [34]. These data serve as a complete set of raw data for scenario prediction and evaluation of the spread of COVID-19 with prevention and control measures.

Specifically, the data used in the infectious disease model included time series data of the daily new cases, deaths, recoveries, and vaccination. Four types of prevention and control measure data were considered to analyze the relationship between the daily new cases and preventative measures: (1) pandemic data: daily new confirmed cases; (2) vaccine data: daily new vaccination data; (3) prevention and control measure data [35] (Table 1): closure and containment measures (school closures, workplace closures, public event cancellations, restrictions on gatherings, public transport closures, stay at home requirements, restrictions on internal movement, and restrictions on international travel), economic measures (income support and debt/contract relief for households), and public health measures (public information campaigns, testing policies, contact tracing, face coverings, and vaccination policies); (4) comprehensive prevention and control data in the form of the stringency index, which is a composite measure based on nine response indicators including school closure, workplace closure, public event cancellations, restrictions on gatherings, stay at home requirements, restrictions on internal movement, restrictions on international travel and public information campaigns. This value was rescaled to a value from 0 to 100 (100 = strictest) [35].

### 2.2. Model

Susceptible–infected–recovered (SIR) models have been widely used in the prediction and analysis of the COVID-19 outbreak as they can explain the relationship among susceptible, infected and recovered cases [36,37]. For example, the SIR model combined with an artificial intelligence approach successfully predicted the COVID-19 epidemic spread in China [37]. Tian et al. used the SIR model to successfully reproduce the first 50 d of COVID-19 epidemic transmission in China [38]. Moreover, the improved SIR model has been used to predict the pandemic in six African countries, and the validations have been consistent with the real pandemic within 20 d of the initial stage [39]. The susceptible–infected–recovered–vaccinated (SIRV) model, which is based on the SIR model, can more accurately forecast the epidemic because it includes the vaccination module [40,41]. However, in the face of shortage of vaccines, reduced vaccine efficacy and presence of mutated viruses (Alpha, Delta, Delta+, Lambda), nonpharmaceutical interventions are expected to play a crucial role in curbing the spread of COVID-19. Owing to these aspects, the predictions of the SIRV model are questionable, and a model considering nonpharmaceutical intervention measures can likely more realistically predict the spread of COVID-19. Therefore, we proposed a new model based on the SIRV with a nonpharmaceutical intervention module to predict the spread of COVID-19 in South and Southeast Asia. The model is formulated as follows:$$\left\{\begin{array}{l} S\left(t\right)=S\left(t-1\right)-\gamma \times V_{d}-\alpha \times S\left(t-1\right)\times \frac{I\left(t-1\right)}{N}-\lambda \times S\left(t-1\right)\\ I\left(t\right)=I\left(t-1\right)+\alpha \times S\left(t-1\right)\times \frac{I\left(t-1\right)}{N}-\beta \times I\left(t-1\right)\\ R\left(t\right)=R\left(t-1\right)+\beta \times I\left(t-1\right)\\ V\left(t\right)=V\left(t-1\right)+\gamma \times V_{d}\\ Q\left(t\right)=Q\left(t-1\right)+\lambda \times S\left(t-1\right) \end{array}\right.$$
where *S* is the susceptible population, *I* is the infected population, *R* is the recovered population (including recovered and dead individuals), *V* is the population immunized by vaccination, and *Q* is the population protected through nonpharmaceutical intervention measures. These state variables constitute the whole population (*N*). *V_d_* denotes the number of daily vaccinations. Generally, South and Southeast Asia have implemented a two-dose vaccination strategy; therefore, *V_d_* actually represents the full vaccination data per day; otherwise, this value represents half of the number of vaccinations per day. Intuitively, a larger number of vaccinated or protected people corresponds to fewer people that can be infected and a more conducive environment to prevent and control the pandemic. The parameters *α*, *β*, and *γ* represent the infection rate, recovery rate and vaccine efficacy rate, respectively. *λ* represents the protection rate (or intervention factor) associated with the nonpharmaceutical intervention measures.

A higher efficacy rate of the vaccine means a larger number of people immunized under the same number of vaccine doses, which is more conducive for prevention and control. Considering the shortage of vaccines, a 70% average efficacy rate of the vaccine [42,43,44] was adopted. A higher protection rate corresponds to a smaller susceptible population and number of infections. To highlight the crucial role of nonpharmaceutical intervention measures, three scenarios were set in terms of the protection rate to enhance the predictability of the improved SIRV model. The three scenarios corresponded to 1.5 times the current protection rate, current protection rate, and 0.5 times the current protection rate.

### 2.3. Parameter Estimation

A dynamic model is characterized by its parameters, and reasonable parameter estimation is crucial for determining the stationary parameters of a model and enhancing its prediction accuracy [45,46,47,48]. However, the parameters involve notable uncertainties owing to the dynamics of infectious disease models due to the pathological characteristics of the virus and human factors [38,49]. Therefore, the optimal estimation of the parameters was considered to enhance the predictability of the infectious disease model [50]. The Metropolis–Hastings (MH) algorithm has been widely used in various fields, including epidemiology [39,51], ecological modeling [52], hydrology [53], and geology [54]. In this study, the MH algorithm was used for sampling the daily-confirmed cases and vaccination data from the multidimensional distribution space of parameters (α, β, λ) to obtain an optimal estimation of the posterior probabilities of the improved SIRV model parameters by constructing the likelihood function.

Using Bayes’ theorem, the posterior distribution of parameter set θ can be described as $P\left(\theta |\Omega \right)\propto P\left(\Omega |\theta \right)P\left(\theta \right)$, where $P\left(\theta \right)$ is the prior parameter distribution, for which the normal distribution was selected. $P\left(\Omega |\theta \right)$, as the likelihood function, reflects the probability of observation set Ω with parameter θ. Under the normal distribution, $P\left(\Omega |\theta \right)=\prod \prod \frac{1}{\sqrt{2\pi }\sigma \left(\theta \right)}e^{-\frac{\varepsilon^{2}\left(\theta \right)}{2\sigma^{2}\left(\theta \right)}}$, where $\varepsilon \left(\theta \right)$ and $\sigma \left(\theta \right)$ represent the deviation and standard deviation between the model simulations and real observations based on parameter, respectively. We set the number of samples and iterations as 10^4^. The existing number of confirmed cases were compared with the number of cases simulated using the improved SIRV model. The acceptance rate of the candidate parameter was obtained according to the likelihood function and a normal distribution until the end of the iteration. A detailed description of the MH algorithm can be found in references [39,51,52].

For the MH parameter algorithm, the range of parameters to be optimized must be specified in advance, followed by sampling and iterating the values in the multidimensional parameter space. In this study, the infection rate and recovery rate of the epidemic were calculated in advance according to the confirmed and recovered cases, respectively. The parameter ranges were set as (0.8α 1.2α) and (0.8β 1.2β). The protection rate range was set as (0 0.2) [39].

### 2.4. Design of Experiments

The daily newly confirmed cases in 19 countries of South and Southeast Asia were analyzed. The relationship between 15 intervention measures (C1–C8, E1–E2, H1–H5) and daily new cases for South and Southeast Asia was considered to identify the measures that were efficient before and after vaccination to explain the rebound of the pandemic and guide future interventions. The changes in the growth rate of daily new cases were analyzed before and after the nodes representing prevention and control measure adoption to identify the shortcomings in the interventions to provide reference for adjusting pandemic prevention policies. Finally, the SIRV model including a module of nonpharmaceutical interventions was used to predict the COVID-19 spread in three scenarios to indicate the role of nonpharmaceutical interventions and evaluate the disease. Specifically, the MH algorithm was incorporated in the improved SIRV model to predict the real pandemic data by considering vaccination and nonpharmaceutical intervention measures to increase the parameter accuracy and enhance the predictability of COVID-19. The model parameters were dynamically adjusted considering the real confirmed cases, recovered cases, and vaccinations in each country. According to the protection rate of nonpharmaceutical interventions, three scenarios (optimistic, neutral, and pessimistic) were set to forecast the trend of the spread of COVID-19 in South and Southeast Asian countries. The following scenarios were set: (1) optimistic scenario, corresponding to 1.5 times the current protection rate; (2) neutral scenario, corresponding to the current protection rate; (3) pessimistic scenario, corresponding to 0.5 times the current protection rate.

## 3. Results and Analysis

The pandemic status in South and Southeast Asia remains severe. Even after the introduction of vaccines, COVID-19 spread rapidly in 19 countries of South and Southeast Asia, except Brunei and Singapore (Figure 1). In particular, the COVID-19 spread in the last 5 months in most countries of South and Southeast Asia was more severe than that in the previous scenarios. This severe phenomenon can be attributed to five reasons. (1) Owing to the institutional infrastructure, cultural characteristics and societal conditions, most South and Southeast Asian countries could not rapidly curb the spread of COVID-19. (2) The spread of COVID-19 in most countries was affected by the rebound of the pandemic in India in the first half of 2021. (3) The vaccination rates in most South and Southeast Asian countries are extremely low. For example, fewer than 50 vaccine doses per 100 people were noted to be available in 12 countries, while the attainment of herd immunity requires at least 120 doses of vaccine per 100 people [15]. (4) The emergence of immune-escape variants could be another challenge. Increased infectivity and shorter incubation period of the mutated virus (Delta, Delta+, and Lambda) led to the rapid spread of COVID-19. (5) Nonpharmaceutical intervention measures have been loosened. The shortage and unfair allocation of vaccines and mutation of the virus have not been rapidly addressed. Therefore, it is necessary to analyze the nonpharmaceutical intervention measures in South and Southeast Asian countries.

### 3.1. Retrospective Analysis of the Impact of Intervention Measures on Daily New Cases

The relationship between the detailed intervention measures and countries that adopted each specific measure (Figure 2; results for Maldives are not presented due to data shortage) shows that most countries adopted effective closure and containment measures before vaccines were available. Economic and public health measures are complementary. Restrictions on international travel, restrictions on gatherings, and public transport closures are ranked highly among all closure and containment measures before vaccination. Debt/contract relief for households is the main economic measure, and contact tracing and face coverings are the main public health measures. Notably, public transport closure and restrictions on internal movement rank highly among all closure and containment measures after vaccination. Face coverings represent the most highly ranked public health measure. The number of countries implementing effective interventions has decreased after the availability of vaccines, and the response intensity of the intervention measures to daily new cases has decreased in most countries. For example, Brunei, Malaysia, and Pakistan have loosened all measures, and Cambodia, the Philippines, and Vietnam have loosened closure and containment measures.

Although the effective intervention measures in each country have been dynamically adjusted with the spread of COVID-19, each country has relaxed the recommended measures following vaccine intervention. However, the daily new cases in most countries of South and Southeast Asia have continued to increase (Figure 1). This phenomenon indicates that loosening nonpharmaceutical interventions under a shortage of vaccines is not conducive to curbing COVID-19. For example, the increasing curves for the Philippines, Thailand, and Vietnam show that it is not suitable to prematurely relax prevention and control measures, and neglect the adoption of effective measures. Although vaccines are key to curbing the epidemic, scientific effective prevention and control measures must be emphasized before herd immunity is reached [55].

### 3.2. Retrospective Analysis of the Growth Rate for Daily New Cases

Evaluation of the growth rate of daily new cases can provide reference for recognizing the spread of the pandemic and adjusting intervention measures. The growth rate of daily new cases (Table 2) was analyzed depending on the adopted closure and containment measures (C1–C8), public health measures (H1–H4) and lockdowns. Statistics indicate that the growth rate of daily new cases decreased significantly after the intervention measures were adopted, with a growth rate of 5.47% before and 1.85% after implementing 12 measures, respectively, and a growth rate of 27.30% before and −0.86% after lockdowns. The growth rate of daily new cases increased when intervention measures were loosened (2.98% and 4.07% before and after loosening measures, respectively). Notably, vaccine intervention failed to significantly reduce the daily growth rates: the rate was 0.67% before vaccination, and 2.44% and 2.05% after 14 and 28 d of vaccination, respectively, indicating that lower vaccination rates and relaxation of intervention measures contributed to the recent spread of COVID-19 in most South and Southeast Asian countries. Specifically, after the implementation of the C1–C8 and H1–H4 measures, the growth rates of daily new cases in Bangladesh, India, Indonesia, Malaysia, Nepal, Pakistan, the Philippines, Sri Lanka, and Thailand decreased, while those in Afghanistan and Myanmar slightly increased. The nonpharmaceutical intervention of lockdowns helped considerably reduce the growth rate of daily new cases. After the initial relaxation of measures, the growth rate of daily new cases in Indonesia, Malaysia, Myanmar, the Philippines, Sri Lanka, Timor, and Vietnam increased, which demonstrates that the relaxation of prevention and control measures exacerbated the spread of COVID-19. In addition, vaccination failed to decrease the growth rate of daily new cases on an ongoing basis. Only in Cambodia and Singapore did the growth rate of new cases continue to decline after vaccination. In Indonesia, Myanmar and Sri Lanka, the growth rate of daily new cases decreased sharply within one month after vaccinations were introduced and continued to decrease one month later. The growth rate of daily new cases in Afghanistan, Bangladesh, India, Malaysia, Pakistan, and Thailand continued to increase. The growth rate of daily new cases in Nepal, the Philippines, and Timor continued to rise within one month after vaccination, but the next month, the growth rate slightly decreased. Generally, the growth rate of daily new cases was expected to significantly reduce by vaccine intervention. In this context, it is necessary to identify the reason for the continued increase in the growth rate of daily new cases after vaccination.

We evaluated the influence of the stringency index, which is a composite measure based on C1–C8 and H1 rescaled to a value from 0 to 100, on the growth rate of daily new cases before and after vaccination (Appendix A, Figure A1). Within a month of vaccination, Indonesia, Cambodia, and Thailand gradually strengthened their stringency index. Myanmar, Bangladesh, Bhutan, and Brunei maintained stringency indices of approximately 70%, 80%, 70%, and 40%, respectively. However, other countries adopted either weaken–strengthen or strengthen–weaken trends for stringency. For example, the stringency index in Nepal decreased to 20% and later increased to nearly 90% on 29 April. India’s stringency index slowly declined, increased to more than 70% on 3 April, and further increased to 80% on 10 May. Malaysia continued to weaken its stringency index until it approached 50%, although this value increased to more than 80% on 12 May.

The weighted average (Appendix B) can reflect the trend in a group of data. The weighted average (Figure 3) of the growth rate of daily new cases in 18 countries (except for Maldives due to data shortage) indicates that the rate decreased after lockdowns and 12 interventions were implemented; however, the rate increased after one or more measures were loosened (after 14 d) and vaccines were introduced. Moreover, Appendix A Figure A1 shows that the growth rate of daily new cases was consistent with the weighted average values in most countries. In addition, a correlation could be observed between the growth rate of daily new cases and relative change in the stringency index. The growth rate of daily new cases increased in Afghanistan, Bangladesh, India, Malaysia, Nepal, Pakistan, the Philippines, Sri Lanka, Thailand, Timor and Vietnam, whereas in other countries, the growth rates decreased after vaccination. The stringency index is related to the growth rate of daily new cases (Appendix A, Figure A1). The growth rate of daily new cases increased in these 11 countries, corresponding to a decrease in the stringency index or a continued maintenance at a low level within a month of vaccination. The growth rates in the other countries decreased as the stringency index increased. Therefore, when the vaccine intervention is implemented at a small scale, the stringency index can be increased to effectively control the spread of COVID-19.

Daily new cases are continuously increasing in Brunei, Laos, Malaysia, Myanmar, the Philippines, Sri Lanka, Thailand, Timor, and Vietnam at present, owing to the reduction in the stringency index in these countries.

### 3.3. Prediction of the Spread of COVID-19 in South and Southeast Asia

The spread of COVID-19 in 8 countries in South Asia (Afghanistan, Bangladesh, Bhutan, India, Maldives, Nepal, Pakistan, and Sri Lanka) and 11 countries in Southeast Asia (Brunei, Cambodia, Indonesia, Laos, Malaysia, Myanmar, Philippines, Singapore, Thailand, Timor, and Vietnam) was predicted (Figure 4) using the improved SIRV model. The findings demonstrated the crucial role of nonpharmaceutical interventions. In the neutral scenario (current protection rate), the daily new cases in Afghanistan, Bangladesh, Bhutan, India, Maldives, Nepal, and Pakistan in South Asia exhibit a gentle downward trend. Afghanistan, India, Nepal, and Pakistan are expected to control daily new cases to within 4, 430, 220, and 20 by the end of 2021, respectively. Bangladesh, Bhutan, and Maldives are expected to control daily new cases to within one on 31 December, 16 September, and 30 October, respectively. However, Sri Lanka exhibits an increasing trend for the number of new cases, and the number of new cases reaches its peak (4200 cases) on 29 August. The daily new cases in Cambodia, Indonesia, Myanmar, Singapore, and Thailand in Southeast Asia exhibit a downward trend. Cambodia and Singapore are expected to control the daily new cases to within 1 on 28 October and 25 October, respectively. Indonesia, Myanmar, and Thailand are expected to control the daily new cases to within 12, 7, and 5 by the end of 2021, respectively. COVID-19 in the other six countries (Brunei, Laos, Malaysia, the Philippines, Timor, and Vietnam) in Southeast Asia is expected to spread, with the peak numbers of new cases (330, 810, 26,300, 31,000, 330, and 20,200) expected to occur on 25 September, 14 October, 12 September, 13 October, 5 September, and 13 October 2021, respectively.

Compared to those in the neutral scenario, the increasing curves of new cases are expected to be sharper in the pessimistic scenario (0.5 times the current protection rate), and the daily new cases are expected to increase earlier than those in the neutral scenarios. For example, daily new cases in Laos are expected to reach 810 on 24 September 2021. Similarly, the decreasing curves are expected to be flatter compared with those in the neutral scenario, and the benchmark numbers of daily new cases are expected to be delayed compared to those in the neutral scenario. For example, the number of new cases in Maldives is expected to reduce to one on 1 November 2021. Additionally, the peak number of infections is expected to be delayed by 5–30 d, and the maximum number of infections is expected to increase in the pessimistic scenario. For example, the peak number of new cases (1150) in Laos is expected to occur on 24 October 2021.

The curves of daily new cases exhibit a contrasting trend in the optimistic scenario (1.5 times the current protection rate) compared with that in the pessimistic scenario. The increasing curves are expected to be flatter, the decreasing curves are expected to be sharper, the number of infection peak dates is expected to advance by 5–20 d, and the maximum number of infections is expected to be smaller than those in the neutral and pessimistic scenarios. For example, the number of new cases in India is expected to reduce to 430 on 30 November 2021, and Laos is expected to witness the peak number of new cases (700) on 7 October 2021, in this scenario.

In summary, the daily new cases are expected to decrease rapidly or increase with a flatter curve with an increasing protection rate. The peak number of infections is expected to occur earlier, and the maximum number of infections is expected to decrease. These findings indicate that proper nonpharmaceutical intervention adoption is the key strategy to alleviate the spread of COVID-19 in this stage.

## 4. Conclusions

Although South and Southeast Asian countries adopted strict prevention and control measures, the spread of COVID-19 in most countries worsened after the introduction of vaccines.

Evaluation of the nonpharmaceutical intervention indicated that most South and Southeast Asian countries loosened the effective measures after the introduction of vaccines. For example, Brunei, Cambodia, the Philippines, and Vietnam completely loosened closures and containment measures. However, loosening the nonpharmaceutical interventions before reaching herd immunity promoted the rebound of the pandemic in most South and Southeast Asian countries following the outbreak in India and mutation of the virus (Alpha, Delta, Delta+, and Lambda).

The time nodes for adopting the intervention measures indicated that strong intervention measures could retard the spread of COVID-19. Statistics indicated that the growth rate of daily new cases decreased significantly after the intervention measures were adopted, with a growth rate of 5.47% before and 1.85% after the implementation of 12 measures and a growth rate of 27.30% before and −0.86% after lockdowns. If COVID-19 is not effectively controlled, the relaxation of prevention and control measures can further aggravate its spread, despite vaccinations. Notably, vaccine intervention failed to significantly reduce the daily growth rates: the rate was 0.67% before vaccination, and 2.44% and 2.05% after 14 and 28 d of vaccination, respectively. This finding highlights that nonpharmaceutical interventions must be strengthened as the primary control measure before herd immunity is attained through vaccination.

Additionally, MH sampling and the SIRV model with a nonpharmaceutical intervention module were used to simulate the spread of COVID-19 in three scenarios (optimistic, neutral and pessimistic) in South and Southeast Asia. The results indicated that the daily new cases are expected to decrease rapidly or increase with a flatter curve with increasing protection rate, and the peak number of infections is expected to occur earlier (5–20 d) with limited cases. For example, India’s new cases are expected to reduce to 430 on 30 November 2021, and Laos is expected to witness the peak number of new cases (700) on 7 October 2021, in the optimistic scenario. These results demonstrate that adopting scientific nonpharmaceutical interventions is the key strategy to alleviate the spread of COVID-19 in this stage.

Vaccines are key to containing the spread of COVID-19. Determination of the herd immunity is based on the efficacy, validity period, and vaccination rate of vaccines. Vaccine administration combined with strict prevention and control measures and big data innovation [50,56] is currently the most efficient tactic for curbing the spread of COVID-19. The continued spread of COVID-19 in South and Southeast Asian countries is a reminder that this pandemic remains a major risk until mass herd immunity is attained. In the long-term, incomplete nondrug prevention and control measures can only delay the deterioration of the pandemic, and the containment of COVID-19 depends on the rate of vaccination.

In particular, Delta-type viruses have a shorter incubation period (2–4 d), strong infectivity and undetermined vaccine effectiveness. With the rapid spread of this mutated virus, prevention and control measures should not be relaxed before mass herd immunity is reached. Emergency-authorized vaccines have been noted to be effective in preventing infections and severe cases, and it is necessary to accelerate vaccination in an appropriate manner and dynamically strengthen nonpharmaceutical intervention measures worldwide.

## Figures and Tables

**Figure 1 healthcare-09-01292-f001:**
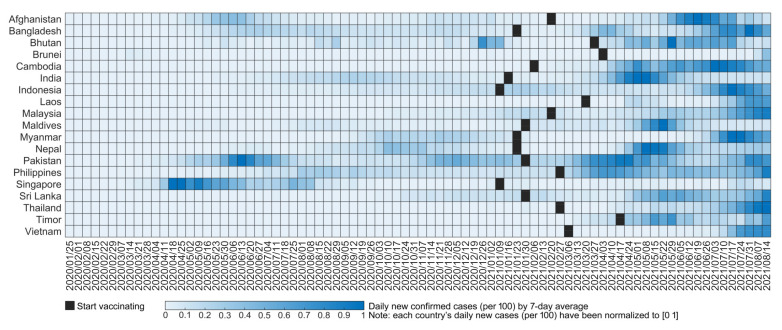
Daily new confirmed cases (per 100) in 19 countries of South and Southeast Asia (the daily new cases (per 100) for each country have been normalized to (0, 1) to clearly show the results).

**Figure 2 healthcare-09-01292-f002:**
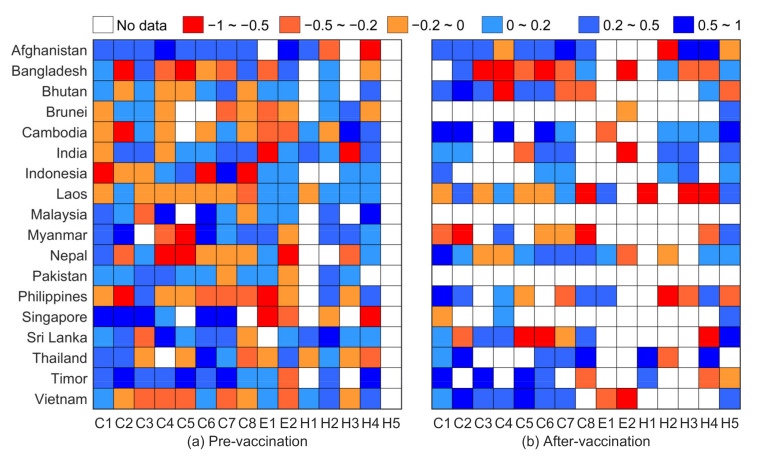
Response intensity of intervention measures to daily new cases before and after vaccine availability (response intensity is based on a correlation analysis; a stronger response corresponds to a larger negative value). Closures and containment measures including school closures (C1), workplace closures (C2), public event cancellations (C3), restrictions on gatherings (C4), public transport closures (C5), stay at home requirements (C6), restrictions on internal movement (C7), and restrictions on international travel (C8). Economic measures included income support (E1) and debt/contract relief for households (E2). Public health measures include public information campaigns (H1), testing policies (H2), contact tracing (H3), face coverings (H4), and vaccination policies (H5).

**Figure 3 healthcare-09-01292-f003:**
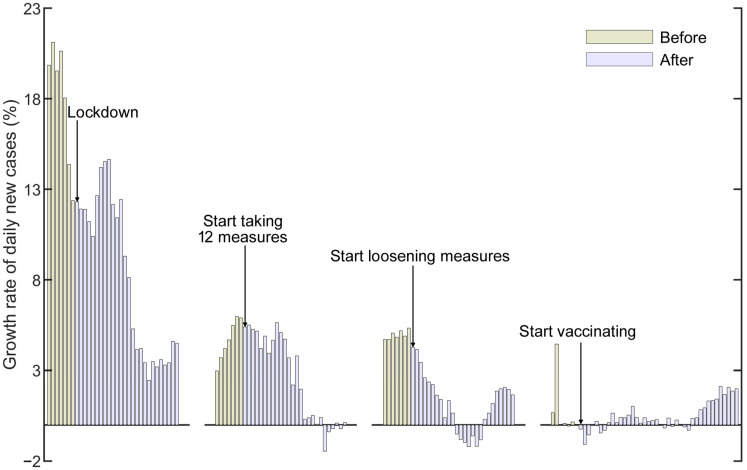
Weighted average growth rate of daily new cases for 18 countries (except for Maldives due to data shortage) in South and Southeast Asia. Pale yellow and light purple represent values before and after the implementation of lockdowns and 12 measures (C1–C8 and H1–H4), loosening of these measures, and introduction of vaccines, respectively.

**Figure 4 healthcare-09-01292-f004:**
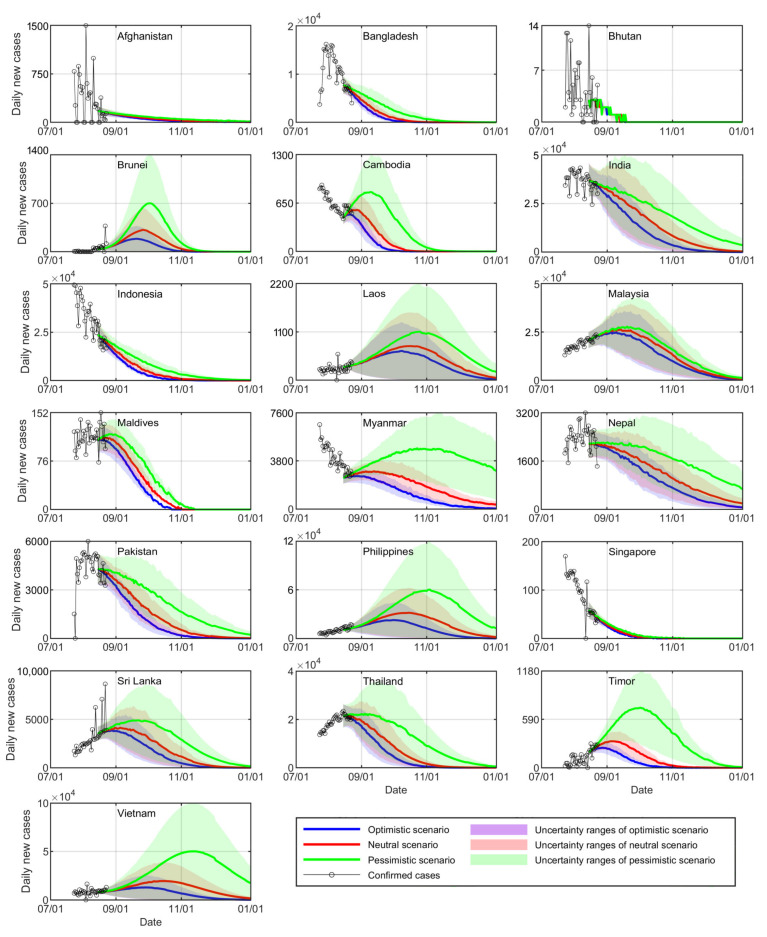
Predicted daily new cases in 19 countries in South and Southeast Asia in three scenarios (optimistic scenario: 1.5 times the current protection rate. Neutral scenario: current protection rate. Pessimistic scenario: 0.5 times the current protection rate).

**Table 1 healthcare-09-01292-t001:** Description of the specific intervention measures.

Prevention and Control Measures	Detailed Measures (Identity Code)	Corresponding Measures (Intensity)
Closures and containment measures	School closures (C1)	Complete closure (3), partial suspension (2), suggested closure (1), no measures (0)
	Workplace closures (C2)	Closure of all nonessential workplaces (3), closure of several workplaces (2), suggested closure (1), no measures (0)
	Public event cancellations (C3)	Mandatory cancellation (2), suggested cancellation (1), no measures (0)
	Restrictions on gatherings (C4)	No more than 10 people (4), no more than 100 people (3), no more than 1000 people (2), limits only on major events (1), no measures (0)
	Public transport closures (C5)	Complete closure (2), suggested closure (1), no measures (0)
	Stay at home requirements (C6)	Home except for special events (3), home except for daily activities (2), no measures (0)
	Restrictions on internal movement (C7)	Regional/local movement restrictions (2), interregional movement restrictions (1), no measures (0)
	Restrictions on international travel (C8)	Closure of all borders (4), closure of selective borders (3), high-risk imports quarantined (2), screenings (1), no measures (0)
Economic measures	Income support (E1)	Government subsidizes more than 50% of wages (2), government subsidizes less than 50% of wages (1), no measures (0)
	Debt/contract relief for households (E2)	Broad relief (2), targeted extensions (1), no measures (0)
Public health measures	Public information campaigns (H1)	Extensive publicity (2), government supervised campaigns (1), no measures (0)
	Testing policies (H2)	Extensive testing (3), symptomatic testing (2), symptomatic testing under specific criteria (1), no measures (0)
	Contact tracing (H3)	Trace all contacts (2), trace limited contacts (1), no measures (0)
	Face coverings (H4)	Always (4), in public spaces (3), social distancing when possible (2), recommended (1), no measures (0)
	Vaccination policies (H5)	Universal availability (5), key groups (essential workers/clinically vulnerable groups/elderly groups) and broad groups/ages (4), key groups only (3), two-thirds of key groups (2), one-third of key groups (1), no measures (0)

**Table 2 healthcare-09-01292-t002:** Growth rate of daily new cases before and after intervention measures (except for Maldives due to data shortage).

Country	Growth Rate of Daily New Cases								
	Before Lockdown	After Lockdown	Before Adopting Measures (C1–C8 and H1–H4)	After Adopting Measures (C1–C8 and H1–H4)	Before Loosening Measures	After Loosening Measures	Before Vaccination	After Vaccination I	After Vaccination II
Afghanistan	20.35%	6.43%	6.28%	6.41%	0.79%	−1.20%	−3.59%	3.73%	6.93%
Bangladesh	21.54%	15.15%	3.94%	1.18%	8.42%	−4.72%	−3.19%	0.53%	5.42%
Bhutan	——	——	——	——	22.00%	4.25%	——	7.25%	5.02%
Brunei	4.89%	−18.30%	——	——	——	——	16.71%	19.23%	−1.31%
Cambodia	−7.40%	0.00%	——	——	——	——	2.87%	1.87%	1.73%
India	20.35%	6.30%	18.33%	4.56%	5.89%	4.01%	−1.33%	−0.95%	2.21%
Indonesia	48.65%	6.77%	7.05%	1.56%	3.89%	4.03%	2.59%	−1.58%	−0.71%
Laos	——	——	——	0.00%	0.00%	0.00%	0.00%	0.00%	0.00%
Malaysia	23.35%	−1.75%	4.46%	1.82%	−0.88%	2.42%	−1.54%	−1.13%	−0.44%
Myanmar	——	——	8.96%	16.23%	5.90%	29.97%	−2.21%	−6.51%	−0.04%
Nepal	——	——	7.15%	3.24%	7.65%	2.69%	−2.87%	−1.08%	−1.32%
Pakistan	36.43%	7.28%	5.36%	−3.36%	5.39%	−4.59%	−2.07%	1.21%	5.10%
Philippines	70.11%	5.52%	0.77%	0.07%	0.07%	1.40%	1.90%	5.30%	0.95%
Singapore	7.38%	−2.32%	——	——	——	——	6.63%	−1.71%	−5.01%
Sri Lanka	66.39%	1.07%	1.26%	−0.10%	−2.20%	14.06%	1.76%	−4.23%	−2.16%
Thailand	38.47%	−7.90%	−2.28%	−8.85%	11.61%	5.17%	−4.79%	4.24%	15.43%
Timor	——	——	——	1.29%	2.15%	4.14%	3.62%	4.93%	1.36%
Vietnam	4.34%	−29.42%	4.34%	——	−25.80%	−0.34%	−3.10%	12.77%	3.80%
Average	27.30%	−0.86%	5.47%	1.85%	2.98%	4.09%	0.67%	2.44%	2.05%

—— No data. Vaccination I: after 14 d of vaccination. Vaccination II: after 28 d of vaccination. Loosening measures means one or more measures have been relaxed.

## Data Availability

The data presented in this study are available upon request from the corresponding author.

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
