# Peer review of "Stringent Nonpharmaceutical Interventions Are Crucial for Curbing COVID-19 Transmission in the Course of Vaccination: A Case Study of South and Southeast Asian Countries"

_healthcare, 2021, doi:10.3390/healthcare9101292_

Round 1
Reviewer 1 Report
Thank you for the opportunity to review this modeling study. Good analysis on the whole. Please see below for my specific comments.
Specific comments:
- As per the journal's guidelines, the abstract should be a maximum of 200 words.
- "... the normalization of the pandemic" - awkward word choice and phrasing, please rephrase.
- More background on the stringency index would be helpful, for example, explaining that this is a composite measure based on nine response indicators including school closures, workplace closures, and travel bans, rescaled to a value from 0 to 100 (100 = strictest).
- What is meant by the term "conditional vaccination"?
- "Asia has become a key region for risk because it holds the most daily new cases, especially in India" - is it really a key region for risk given the high number of daily cases in the Western world as well? Or is it because of the emergence of the highly-transmissible delta variant?
- Are the susceptible–infected–recovered (SIR) models validated? Some evidence should be referenced and cited.
- It is important to mention that current evidence suggests the vaccines likely do not affect viral shedding or transmission dynamics.
- In terms of pre-emptive strategies, state policies mandating public or community use of face masks or covers and adherence to social distancing are perhaps the most effective strategies available based on current data, and they should be rightly discussed (citation: pubmed.ncbi.nlm.nih.gov/32543923; ncbi.nlm.nih.gov/pmc/articles/PMC7316449).
- If policies vary at the subnational level, the stringency index is shown as the response level of the strictest sub-region, this may muddle the analysis.
- In terms of technology and COVID-19, the key here is rigorous contact tracing, which was not well-expressed by the authors. Compliance and data and personal privacy issues are also major issues limiting the uptake, use, and effectiveness of these technologies.
- The premise of this article needs to be more rigorously defended with real-life data and case studies, which are already available in the form of countries and states that have not mandated universal masking.
Reviewer 2 Report
Many thanks for giving me the opportunity to read this paper. Overall, the paper is sound & the results are of interest. I particularly enjoy that you confront vaccination with other measures, via the stringency index. I have however several comments:
1) First, the focus on SouthEast Asia is not well motivated. Why should we have a look on a part of Asia instead of Asia as a whole or Europe or the World ? As you are using a global dataset (Oxford Tracker), & did not have to hand collect the data, we can wonder why you use such a narrow focus.
2) You focus on the stringency index, which is criticable. Evidence shows that different measures have different impacts. For example, Haug et al. (2020) or An et al. (2021) show that some measures are more efficient than others. Cheng et al. (2020) also show that different measures do not have the same score of stringency as discussed also in Porcher (2020). Indeed, as discussed in Hale et al. (2021) or Porcher and Renault (2021), mobility might be more or less affected by some non-pharmaceutical interventions. Intuitively, masks allow people to be mobile while limiting contaminations at the same time; domestic lockdown are harsh measures. Why don't you focus on some key NPIs ? And how the results would change if you use only 5 or 6 key NPIs as discussed in Flaxman et al. (2020) ? I suggest that you rerun the experiment considering the measures in a separated, rather than aggregated, form.
3) What is the tradeoff between vaccination and stringency? In descriptive data, stringency increases right after the beginning of the vaccination. If you looked at some specific measures, e.g. lockdown, the results could be more interesting: are we dropping the harsher measures after the vaccination campaigns ?
4) Are there specific government or cultural characteristics that could explain the different successes of policies within Asia ?
I would suggest the authors to add the following references:
- An, Brian Y., and Shui-Yan Tang. 2020. Lessons from COVID-19 responses in East Asia: Institutional infrastructure and enduring policy instruments. The American Review of Public Administration 50(6-7): 790-800. https://doi.org/10.1177/0275074020943707.
- An, B, Porcher, S., Tang S-Y. and E. Kim. Policy Design for COVID-19: Worldwide Evidence on the Efficacies of Early Mask Mandates and Other Policy Interventions. Public Administration Review, in press. doi: 10.1002/puar.13426.
- Flaxman, S. et al. Estimating the effects of non-pharmaceutical interventions on COVID-19 in Europe. Nature 584, 257–261 (2020
- Haug, N., Geyrhofer, L., Londei, A. et al. Ranking the effectiveness of worldwide COVID-19 government interventions. Nat Hum Behav 4, 1303–1312 (2020).
- Cheng, C., Barceló, J., Hartnett, A. S., Kubinec, R. & Messerschmidt, L. COVID-19 government response event dataset (CoronaNet v.1.0). Nat. Hum. Behav. 4, 756–768 (2020).
- Porcher, S. Response2covid19, a dataset of governments’ responses to COVID-19 all around the world. Sci Data 7, 423 (2020). https://doi.org/10.1038/s41597-020-00757-y
- Porcher, Simon and Thomas Renault. 2021. Social distancing beliefs and human mobility: Evidence from Twitter. PLOS ONE. https://doi.org/10.1371/journal.pone.0246949.
Round 2
Reviewer 1 Report
Thank you for the revisions made.
Suggest a close edit for grammar and language issues, still rather rampant throughout the manuscript.
Reviewer 2 Report
The authors responded to my comments and the manuscript is now improved, with a clearer scope and results presented in a more intelligible way. The context of Southern Asia is also well introduced.
On page 10, there is an empty table that should be removed.
This manuscript is a resubmission of an earlier submission. The following is a list of the peer review reports and author responses from that submission.